# Impact of Plasma-Activated Water Treatment on Quality and Shelf-Life of Fresh Spinach Leaves Evaluated by Comprehensive Metabolomic Analysis

**DOI:** 10.3390/foods10123067

**Published:** 2021-12-09

**Authors:** Oscar Daniel Rangel-Huerta, Lada Ivanova, Silvio Uhlig, Morten Sivertsvik, Izumi Sone, Estefanía Noriega Fernández, Christiane Kruse Fæste

**Affiliations:** 1Section for Chemistry and Toxinology, Norwegian Veterinary Institute, P.O. Box 64, N-1431 Ås, Norway; lada.ivanova@vetinst.no (L.I.); silvio.uhlig@vetinst.no (S.U.); christiane.faste@vetinst.no (C.K.F.); 2Nofima AS, Department of Processing Technology, Richard Johnsens Gate 4, 4021 Stavanger, Norway; morten.sivertsvik@nofima.no (M.S.); Izumi.sone@nofima.no (I.S.); estefania.noriegafernandez@efsa.europa.eu (E.N.F.); 3European Food Safety Authority (EFSA), Via Carlo Magno 1A, 43126 Parma, Italy

**Keywords:** food processing, food quality, metabolomics, plasma-activated water, senescence, spinach leaves

## Abstract

Fresh baby spinach leaves are popular in salads and are sold as chilled and plastic-packed products. They are of high nutritional value but very perishable due to microbial contamination and enzymatic browning resulting from leaf senescence. Therefore, innovative food processing methods such as plasma-activated water (PAW) treatment are being explored regarding their applicability for ensuring food safety. PAW’s impact on food quality and shelf-life extension has, however, not been investigated extensively in vegetables so far. In the present study, a comprehensive metabolomic analysis was performed to determine possible changes in the metabolite contents of spinach leaves stored in a refrigerated state for eight days. Liquid chromatography high-resolution mass spectrometry, followed by stringent biostatistics, was used to compare the metabolomes in control, tap-water-rinsed or PAW-rinsed samples. No significant differences were discernible between the treatment groups at the beginning or end of the storage period. The observed loss of nutrients and activation of catabolic pathways were characteristic of a transition into the senescent state. Nonetheless, the presence of several polyphenolic antioxidants and γ-linolenic acid in the PAW-treated leaves indicated a significant increase in stress resistance and health-promoting antioxidant capacity in the sample. Furthermore, the enhancement of carbohydrate-related metabolisms indicated a delay in the senescence development. These findings demonstrated the potential of PAW to benefit food quality and the shelf-life of fresh spinach leaves.

## 1. Introduction

Spinach (*Spinacia oleracea* L.) has a high content of vitamins, antioxidants, minerals, fibres and proteins [1,2]. The vegetable is traditionally eaten cooked, but more recently, the use of fresh spinach in salads has increased considerably. Thus, nowadays, food retailers offer not only frozen bags with pre-cooked spinach or raw spinach heads for home cookery but also refrigerated bags with washed, ready-to-eat baby spinach leaves. The consumption of raw vegetables is advantageous regarding the preservation of their nutritional value. At the same time, the maintenance of food safety is of much concern due to microbial contamination [3]. Leafy greens are typically infected by pathogenic bacteria such as *Salmonella* spp., *Shigella* spp., *Listeria monocytogenes*, *Yersinia enterocolitica* and *Escherichia coli*, which can colonise large surface areas and internalise through the pores into the leaves. Several foodborne outbreaks from contaminated spinach in European countries and in North America have caused health concerns and shown the importance of managing the risks connected with the consumption of fresh produce [4,5]. Food processing methods that efficiently reduce the bacterial load and permit storage of the chilled product for several days are therefore in demand.

Previously used chemical disinfection or irradiation techniques are no longer authorised or appreciated by consumers, and therefore novel food technologies such as treatment with plasma-activated water (PAW) are being explored [6,7]. PAW is produced by the reaction of non-thermal (cold) plasma with water, leading to the generation of a high diversity of reactive oxygen species and reactive nitrogen species with disinfecting properties [8]. The application of this heat-free technology is especially advantageous for fresh fruits and vegetables since it preserves valuable vitamins, keeps the product intact and allows it to retain its colour. Thus, PAW is considered an attractive food preservation method not only with regard to food safety but also with regard to food quality and sensory characteristics [7,8].

The applicability of PAW for the disinfection of baby spinach leaves was recently studied by food technologists in our group [9]. By optimising the plasma operating conditions, a significant drop in bacterial loads during eight days of chilled storage was achieved, while conserving the food’s appearance. While these results showed that PAW was suitable for ensuring product safety and attractiveness, further research is needed in relation to the impact of PAW on food quality.

The detachment of leaves leads to the development of senescence, which is accompanied by the loss of valuable nutrients and enzymatic browning [10,11,12]. Significant changes in the complex chemical compositions of food materials resulting from the application of processing technologies can alter their nutrient contents and sensory characteristics, thus leading to quality loss. Modern measurement methods enabling the broad analytical coverage of food constituents are therefore valuable tools for understanding the impact of these technologies on food. In that sense, the simultaneous determination of the molecules with molecular weights below 1500 Da in a food sample, typically including amino acids, carbohydrates, lipids and phenol derivatives, i.e., the metabolome, has become possible using liquid chromatography high-resolution mass spectrometry (LC-HRMS) [13]. Food metabolomics has the power to differentiate between metabolite profiles in differently treated food products and can thereby provide information on quality aspects. However, metabolic analysis generates extensive datasets, which must undergo a sufficiently discriminant statistical evaluation to ensure the detection of relevant features and avoid over-interpretation of the results [14].

In the present study, we have therefore applied MS-based metabolomics with stringent downstream data processing to reveal potential shifts in the spinach metabolome that could indicate an impact of the PAW treatment on the quality and shelf-life of the fresh leaves.

## 2. Materials and Methods

### 2.1. Reagents

LC-MS-grade water and acetonitrile (MeCN) were from Fisher Scientific (Oslo, Norway), high-performance liquid chromatography (HPLC)-grade methanol (MeOH) from ROMIL (Cambridge, UK) and ammonium carbonate from Fluka (Steinheim, Germany).

### 2.2. Raw Materials

Freshly cut, unwashed baby spinach leaves (*Spinacea oleracea*) were kindly provided in bulk by a local wholesaler (Oslo, Norway) under refrigerated conditions. Immediately after reception, the spinach leaves were sorted by size, and the stems of medium-sized leaves were removed with a sterile scalpel. Eight randomly chosen leaves per sample (5.0 ± 0.2 g) were placed into a sterile glass jar and closed with an aluminium lid. The samples were directly subjected to their respective treatments (Figure 1).

### 2.3. Sample Treatments and Experimental Design

The spinach leaves were processed using three different methods, producing untreated controls (C), tap-water-rinsed (TAP) samples and PAW-rinsed samples (Figure 1). The glass jars containing 5 g of spinach leaves were filled with either 100 mL of TAP or PAW and shaken for 5 min at 20 rpm and 4 °C in a Heidolph Reax 2 overhead shaker (Heidolph Instruments, Schwabach, Germany). Subsequently, samples were recovered using a sieve, and excess liquid was removed with a salad spinner for 3 min. PAW- and TAP-rinsed samples, as well as untreated C samples, were air-packed (Webomatic SuperMax, Bochum, Germany) in 80 µm thick standard sous vide plastic bags (Arne B. Corneliussen AS, Oslo, Norway) and immediately stored. Three replicates per treatment were directly frozen at −80 °C after processing, while three additional replicates per treatment were stored for 8 days at 4 °C (imitating the in-shop conditions) and then frozen at −80 °C prior to analysis, resulting in a total of six sample groups: C and 8-days-stored C (C-S), TAP and TAP-S, and PAW and PAW-S (Figure 1).

### 2.4. PAW Generation and Characterisation

PAW was produced from tap water in a cold plasma (CP) generation system (Phenomenal Aire, Brady Services, Greensboro, NC, USA) 3 days before the spinach-leaves experiment [9]. In brief, a CP system consisting of powered and ground electrodes with a 1 mm thick quartz disc between them was set up to generate a surface barrier discharge with 18 kHz frequency at the lid of the treatment chamber (144 cm^2^ total discharge area). The gap between the liquid surface (100 mL) and the electrode was 44.8 mm (3.2 mm water column). PAW generated with an activation time of 30 min and a plasma power of 36 W had the lowest pH, the highest level of reactive oxygen and nitrogen species, the best oxidation–reduction potential and the longest storage stability at different temperatures. The system operated at atmospheric pressure, with air as the plasma-inducing gas.

The concentrations of nitrates, nitrites and hydrogen peroxide in PAW and TAP were determined with standard spectrophotometric methods as previously described [9] (Shimadzu UVmini-1240 UV–Vis, Shimadzu, Tokyo, Japan). A Spectroquant^®^ test kit #109713 (Merck, Oslo, Norway), according to DIN 38405-9, was used for the quantification of nitrates. Nitrite levels were analysed using the Griess method (according to DIN EN 26777) [15]. Hydrogen peroxide was specified via the titanium sulphate colourimetric method [16]. The pH and ORP values in PAW and TAP were measured with a pH/ion meter (SevenGo pro, Mettler Toledo, Oslo, Norway) (Table 1).

### 2.5. Untargeted Metabolomics Analysis by Liquid Chromatography High-Resolution Mass Spectrometry (LC-HRMS)

#### 2.5.1. Sample Extraction

Samples were dehydrated using liquid nitrogen, weighed and powdered in a mortar. Cold MeOH/H_2_O (50/50, *v*/*v*) was added and mixed using a ThermoMixer (Eppendorf AG, Hamburg, Germany) for 15 min at 4 °C. After centrifugation at 10,000 rpm for 5 min, the supernatant was collected in two separate vials and stored at −80 °C until analysis. A quality control sample (QC) was prepared by pooling 20 µL aliquots of all samples included in the study. The QC sample was mixed and divided into several vials that were interspersed into the analytical sequence.

#### 2.5.2. Untargeted LC-HRMS and LC-HRMS/MS

Aliquots (100 µL) of the spinach extracts were transferred to chromatography vials, which were randomly placed into the autosampler tray of the LC-HRMS and kept at a controlled temperature. The pooled QC sample was run six times, at the beginning and as every sixth sample throughout the entire LC-HRMS experiment.

LC-HRMS analysis was performed using a Q Exactive™ Hybrid Quadrupole-Orbitrap mass spectrometer equipped with a heated electrospray ion source (HESI-II) and coupled to a ultra-high-performance liquid chromatography (UHPLC) Vanquish Horizon system (Thermo Fisher Scientific, San Jose, CA, USA) with a temperature-controlled autosampler.

Chromatographic separation of analytes was achieved by either of the following two approaches:(1)Reverse-phase (RP) chromatography using a Hypersil GOLD aQ (Thermo Fisher Scientific, San Jose, CA, USA; 100 × 2.1, 1.9 µm) column was performed by eluting the column with a mobile phase consisting of water + 0.1% formic acid (FA) (A) and MeOH + 0.1% FA (B). The elution proceeded isocratically at a constant flow rate of 0.4 mL/min for 0.5 min with 100% A, followed by linear gradient elution to 95% B in 4.5 min, which was retained until 12.5 min. After flushing the column with 95% B for 2 min, it was returned to the starting conditions and equilibrated for 2.5 min.(2)Hydrophilic interaction chromatography (HILIC) was performed using a zwitterionic SeQuant ZIC-pHILIC column (Merck, Darmstadt, Germany; 150 × 4.6 mm, 5 µm). The column was eluted with a mobile phase consisting of 20 mM ammonium carbonate (A, pH 8.3) and MeCN (B). Elution proceeded isocratically at a constant flow rate of 0.3 mL/min for 1 min with 80% B, followed by linear gradient elution to 20% B in 29 min. Subsequently, the column was flushed with 8% B for 5 min, and then returned to the starting conditions and equilibrated for 9 min.

For both separation methods, the HRMS instrument was run in full-scan positive and negative ion mode using fast polarity switching in the mass-to-charge (*m*/*z*) range 58 to 870. The HESI-II interface was operated at 300 °C. The spray voltage was 2.8 and 3.2 keV (positive and negative mode, respectively), the ion transfer capillary temperature was 280 °C, the sheath and auxiliary gas flow rates were 35 and 10 units, respectively, and the S-lens RF level was 55%. The automated gain control (AGC) target was set to 5 × 10^5,^ and the maximum injection time (IT) was set to 250 ms. A mass resolution of 75,000 full width half maximum (FWHM) at *m*/*z* 200 was used. All analyses were performed without lock mass. Xcalibur software (version 2.3) was used for instrument control and LC-HRMS data acquisition.

Additionally, a set of LC-HRMS/MS data files were acquired for the QC sample using data-dependent MS/MS mode (DDA). The conditions were as follows: full MS/MS fragmentation scans of the top 5 most intense MS ions were performed in the mass range *m*/*z* 58 to *m*/*z* 870 with a mass resolution of 17,500 for product ion detection. The fragmentation was performed by applying three different collision energies (HCD 15, 35 and 65) in separate runs, independently for each ionisation mode.

#### 2.5.3. Processing and QC of Metabolomics Data

All data were pre-processed in MS-DIAL (v4.60) [17], applying specific parameters for both ionisation modes of each LC method (Appendix A). Four data matrices of detected metabolic features were generated, two for each LC method, which contained retention times (RTs), *m*/*z* values and peak areas normalised by the Locally Weighted Scatterplot Smoothing (LOWESS) function. The LOWESS function considers the injection order and the QC repeats across the run for the normalisation.

Each pair of datasets was subsequently processed using the R-based package MS-CleanR [18], with the application of the following filters: blank signal subtraction (minimum blank ratio set to 0.8), background ion drift removal and determination of relative standard deviation thresholds (RSD, set to 20) based on sample class and relative mass defect (RMD) window filtering (set by default to 50–3000 ppm). As a second step, the data obtained for the two ionisation modes containing MS/MS were merged and clustered according to the MS-DIAL Peak Character Estimation algorithm, followed by parental signal extraction applying multi-level optimisation of the modularity algorithm. The maximum mass difference selected for feature relationships detection was established at 0.005 Da, with a maximum RT difference of 0.05 min. The Pearson correlation links were considered only when the correlation coefficient was ≥0.9 and statistically significant (α = 0.05). During the processing in MS-CleanR, the molecular formulae and a preliminary in silico annotation using MS-FINDER were assigned to the clusters, considering a 5 ppm error for MS data and a 10 ppm error for MS/MS data.

### 2.6. Statistical Analysis

The normalised data were Pareto-scaled in SIMCA (v16; Sartorius Stedim Biotech, Umeå, Sweden) for multivariate analyses using unsupervised (principal component analysis, PCA) and supervised (orthogonal partial least-squares discriminant analysis, OPLS-DA) models. PCA was used to assess the quality of the metabolomics data based on the QCs, to detect potential outliers and to identify clustering patterns. OPLS-DA models were built to identify metabolite patterns and specific features discriminating between the different time points or processing methods. Seven-round cross-validation was applied in the OPLS-DA modelling. Cross-validation of the residuals using ANOVA (CV-ANOVA) was performed to evaluate the reliability of the models, and *p*-values ≤ 0.05 were considered significant.

The most discriminant variables across significant models were extracted comparing the *p*(corr) (a vector representing the correlation and hence the reliability of the data for the processing method Y) and *p*(1) (a vector indicating the modelled covariation) values from each model. For this purpose, a cut-off combining a *p*(corr) ≥ |0.50| and *p*(1) ≥ |0.05| was considered significant. This approach facilitated the extraction of relevant metabolites related to a specific model and allowed the identification of shared features between two or more models with either shared or inverse trends.

### 2.7. Annotation

The annotation process of the most relevant metabolites began during the pre-processing in MS-DIAL by matching the information of the HRMS/MS data obtained for the QC sample to public curated spectral libraries. Annotations were either obtained automatically by matching the measured mass spectra to available MS/MS spectrum repositories, comparing mass accuracies, RTs and isotope patterns [19], or manually curated. Spectral similarity scores between measured features and reference metabolites were determined in MS-DIAL by utilising the combined values of the dot product and reverse dot product, which are both based on relative sums of peak abundances, and the matched fragments ratio. For the identification of metabolites, we set a cut-off of 850 for spectral similarity (the maximum value is 1000). A second step was executed during the pre-processing in MS-CleanR, where formulae and tentative structures were calculated in silico. Furthermore, when the relevant features were part of a cluster (i.e., different types of ions that were related to the same molecule), each feature within the cluster was reviewed to determine its identity (i.e., adduct, neutral loss, fragment, isomer, etc.). For the manually curated annotation of the remaining features with unknown identity, we used the automated class assignment and the ontology prediction tool CANOPUS [20], which is an integral part of the SIRIUS software (v.4.8; available online). This does not require specific configuration settings for systematic chemical classification as it is based on the automated determination of metabolites by ClassyFire [21], based on fragmentation spectra. The use of CANOPUS allowed the biological interpretation of observed metabolite profile changes to be extended without the need to annotate all discriminant metabolites.

#### 2.7.1. Level of Identification

In addition to classifying and annotating each treatment-discriminant metabolite, we also included the level of identification, following the suggested guidelines for metabolomics studies [22]. Level 1 corresponds to the unambiguous identification of a metabolite by matching it to a reference compound with at least two independent and orthogonal properties, e.g., RT and *m*/*z*. Level 2 describes the putative identification of metabolites based on spectral similarities between the HRMS/MS fragmentation data and those of spectral libraries. Level 3 refers to metabolite classes that are tentatively characterised by spectral similarity to published HRMS/MS fragmentation data. Level 4 indicates unidentified or unclassified metabolites that are still differentiable because of their specific spectral data.

#### 2.7.2. MS Peaks to Pathway Analysis

Taking full advantage of the HRMS data generated, we included a functional analysis of the detected metabolites in our data interpretation procedure using the *mummichog* software [23], which is part of the MetaboAnalyst 5.0 [24] platform. Conforming to the limitations of the algorithm, we processed the results obtained by RPLC- and HILIC-HRMS (with merged ionisation modes) separately for C, TAP and PAW samples, giving six datasets in total. Samples at D1 and D8 were compared for each processing method using a paired *t*-test performed with R software, which delivered the *p*-values used as input parameters for the pathway analysis. The same settings were used for all samples throughout the analysis, particularly the mass tolerance of 5 ppm, which ensured that only primary ions were included since the program accepts only *m*/*z* matches of primary ions as valid representatives of the respective metabolites. The primary ions considered were [M + H]^+^, [M + Na]^+^, [M − H_2_O + H]^+^, [M − H]^−^, [M − H_2_O − H]^−^ and [M − 2H]^2−^. Furthermore, the *p*-value cut-off was set at 0.01, and the RTs were included as descriptors. The *Arabidopsis thaliana* (thale cress) database of metabolites, which is the only one available for dicotyledonous plants in *mummichog*, was used as a reference, and only pathways defined by the Kyoto Encyclopedia of Genes and Genomes (KEGG) analysis [25] with at least three entries were included. Pathways with an EASE score ≤ 0.05 were considered to be significantly enriched. The EASE score is a conservative version of Fisher’s exact test and was adopted to increase the robustness of the analysis. 

## 3. Results

### 3.1. Quality Assessment of LC-HRMS Datasets

The metabolomics analyses and pre-processing with MS-DIAL resulted in the detection of 5591 features (3731 in positive and 1860 in negative ionisation mode) using RPLC-HRMS and 6397 features (3990 in positive and 2407 in negative ionisation mode) using HILIC-HRMS. The data were normalised by applying the LOWESS function, with consideration of the analytical variation as determined from the results of the QCs. After this quality assessment, validated features obtained by either ionisation mode were merged, resulting in one dataset containing 981 compounds. A PCA model (Figure 2) was established for initial results inspection. The QCs alignment in the centre of the PCA plots reflected the correct functioning of the LC-HRMS throughout the analytical run for both methods. QC samples were not included in the subsequent data analysis, but none of the spinach samples was removed as an outlier.

Both the immediately deep-frozen samples and the samples stored under refrigeration for 8 days were included in the PCA model. Interestingly, consideration of the storage period led to a clear separation between stored and not-stored samples for all three processing methods. The samples stored at 4 °C for 8 days before freezing, C-S, TAP-S and PAW-S (marked 8 in Figure 2), were grouped in the two sectors on the right side of the PCA scores plot, whereas the samples stored immediately (marked 1) were clustered separately in the two left sectors (Figure 2). The PCA model was valid in accordance with the determined scores for the total explained variance (R^2^X > 0.6) and predictive ability (Q^2^ > 0.25) (Table 2).

### 3.2. Comparison of Processing Methods at Baseline (D1)

The multivariate OPLS-DA comparison of C, TAP and PAW spinach samples, which were immediately deep-frozen at day 1 (D1) and measured by RPLC- and HILIC-HRMS, resulted in negative Q^2^ values, indicating that the model lacked predictability (Table 2). This was also confirmed by additional OPLS-DA for independent pairwise comparisons, resulting in invalid models for C vs. TAP, C vs. PAW and TAP vs. PAW (data not shown). It was thus concluded that there were no discernible differences between the metabolite profiles of C, TAP- and PAW-treated spinach samples at D1.

### 3.3. Comparison of Processing Methods after 8 Days of Refrigerated Storage (D8)

The OPLS-DA comparison of the differently processed spinach samples stored for 8 days under refrigeration conditions (D8) was not significant (Table 2). Additional OPLS-DA for the pairwise comparisons C-S vs. TAP-S, C-S vs. PAW-S and TAP-S vs. PAW-S were not significant either (data not shown), confirming that the metabolite composition of the spinach leaves subjected to 8-day refrigerated storage were very similar. Thus, potential differences are likely to be too subtle to result in statistical models that could differentiate between the samples.

### 3.4. Comparison of Stored (D8) to Baseline (D1) Samples for Each Processing Method

The effect of the different processing methods over time (D1 vs. D8) was investigated by creating three OPLS-DA models: C vs. C-S, TAP vs. TAP-S and PAW vs. PAW-S. All models were statistically significant with CV-ANOVA (*p*-values < 0.05) (Table 2), which indicated a significant change in the metabolite profiles of the spinach leaves after storage at 4 °C for 8 days. As we did not observe significant differences at D1 between the differently processed samples (see Section 3.2), we were able to consider the baseline metabolite profiles as a common reference and to compare the results of the three D1 vs. D8 OPLS-DA models for C, TAP and PAW by using the respective *p*(corr) and *p*(1) values of the annotated metabolites determined with either RPLC- or HILIC-HRMS (Table 3). Thus, the storage-dependent changes in the metabolite profiles could be grouped into three categories: shared, inverse and unique changes, in accordance with the signs of the values. The direction of change in the metabolite profiles is shared when the *p*(corr) values for the three processing methods have the same sign. It is inverse when the signs are opposed, and it is unique when the variable is significant exclusively for one of the treatments.

**Changes shared by all three processing methods**. The comparison of the *p*(corr) values revealed that the concentrations of 12 metabolites increased (7 detected by RPLC and 4 by HILIC) and that those of 12 metabolites decreased (4 detected by RPLC and 8 by HILIC) significantly in C, TAP and PAW samples (Table 3) after storage for 8 days at 4 °C. Among the metabolites with increased levels at D8, we were able to putatively annotate phenylalanine (Figure 3a), 2-isopropylmalic acid and tyrosine, which were detectable after analysis in both ionisation modes and under both LC conditions. Moreover, we detected succinate, threonine and threonate (detected by HILIC in both ionisation modes) and asparagine (detected by HILIC in the positive mode). Tryptophan was measured as a cluster consisting of 16 metabolic features, including the protonated molecule (*m*/*z* 205.0968) as the principal feature and the deprotonated molecule (*m*/*z* 203.0823). Metabolites with decreased levels at D8 included a cluster with *m*/*z* 245.0764 containing pyrimidine nucleosides (uridine, adenosine 3’,5’-cyclic monophosphate and guanosine cyclic monophosphate) and a cluster with *m*/*z* 268.1030 containing purine nucleosides (guanosine and adenosine). Furthermore, a large cluster of 24 features related to *m*/*z* 374.1438 was revealed after closer inspection to comprise metabolites classified as disaccharides, coumaric acid and derivatives, and several phenolic glycosides, as well as the ammoniated form of 1-O-feruloylglucose. In addition, we tentatively annotated pyridoxal (Figure 3b) (detected by both LC modes), spermidine, choline, aspartate and a glucuronic-acid-containing malate.

**Changes shared by two processing methods.** In TAP and PAW samples, the levels of two metabolites increased significantly from D1 to D8. They were tentatively annotated as norleucine and succinic anhydride. The latter, detected by RPLC at *m*/*z* 101.0230, was only just statistically significant for the C samples. This metabolite, which is a dehydration product of succinate, was measured by HILIC with an increasing trend for all processing methods and thus probably belonged in the “shared by all processing methods” category. Four metabolites, two detected by HILIC and two by RPLC, decreased during storage. One metabolic feature was putatively annotated as nicotinamide (*m*/*z* 123.055), while the other three were annotated at class level as amino acid (related to isoleucine), guanidine and fatty amide (Table 3).

In the C and TAP samples, the three metabolites with increased concentrations at D8 were determinable by CANOPUS only at the chemical class level. They corresponded to two α-amino acids (one of them leucine-related) and a bilirubin-related metabolic feature (Table 3). Moreover, four metabolites were decreased after storage. They were designated as one unknown feature (*m*/*z* 368.424), a malic-acid-related cluster (*m*/*z* 133.014), a benzoic-acid-related cluster containing 19 features (*m*/*z* 279.159) and a disaccharide (*m*/*z* 360.148).

Storage-dependent metabolite changes identified only in C and PAW samples included two increased and two decreased features (all measured by HILIC-HRMS). Those showing a higher level at D8 were tentatively annotated as a histidine-related metabolite and a cinnamic acid amide, whereas those where lower levels were observed were the tricarboxylic acids citrate and aconitate. The same features also showed similar but not significant trends in the TAP samples, so they probably belonged to the “shared by all processing methods” category.

**Unique changes.** Only one metabolite, a leucine-related amino acid detected by RPLC-HRMS, was found to change uniquely in the C samples during the storage period (Table 3). Four metabolites were typical of the TAP samples (two identified by each LC method), of which two were annotated at class level as alpha-amino acids and two were unknowns. In the PAW samples, two metabolites were annotated at subclass level as cinnamic acid amide and an α-hydroxy acid derivative, which according to its mass spectrum could be associated with methoxytyramine, a metabolite also identified among the features with an inverse trend (Table 3). Six metabolites decreased uniquely in the PAW samples. They were provisionally annotated as cytidine, adenine, glycerophosphocholine-like (possibly γ-linoleic acid), lysophosphatidylcholine-like (tentatively LysoPC 16:0), a flavonoid-7-O-glucuronide (tentatively 5,3′,4′-trihydroxy-3-methoxy-6,7-methylenedioxyflavone 4′-glucuronide) and an unknown feature.

**Inverse changes.** Several metabolites were identified that showed deviating concentration changes between the three processing methods. We were able to annotate 3-methoxytyramine, which decreased significantly from D1 to D8 in C and TAP samples but showed a significant increase in PAW samples (Table 2) with level 2 probability based on good congruency of the measured and reference MS/MS spectra (Figure 3c). The same trend was observable for tyramine and a cluster of 12 metabolic features containing adducts, fragments and other derivatives of 2-(*p*-coumaroyl)malate and other coumaric acid esters (Table 3). Glutamine decreased in C and increased in PAW samples, whereas valine and arginine significantly increased in C and significantly decreased in PAW samples. A metabolite putatively annotated as a fatty amide, however, was the only one that increased in TAP samples and decreased in C and PAW samples.

### 3.5. Storage- and Processing-Induced Changes in Metabolic Pathways

MS Peaks to Pathway analysis revealed the connection of the observed changes in the spinach metabolome from processing and during storage to specific metabolic pathways (KEGG) (Figure 4, Appendix A).

We observed that the degradation, metabolism and biosynthesis of several amino acids were enriched differently in the experimental groups. Furthermore, it was remarkable that several sugar metabolism pathways were affected in PAW-rinsed samples (Figure 4, Table 4).

## 4. Discussion

In 2011, the FAO reported that around one third of all produced food was annually lost or wasted. One target defined in the United Nations’ Sustainable Development Goals (Target 12.3) [26] aims to halve the global per capita food waste at the retail and consumer level and to reduce food losses along production and supply chains by 2030. The extension of the shelf-life is thus an important contribution to reducing food waste; however, the improvement of food safety should not negatively impact the quality and sensory properties of food products, otherwise consumer acceptance might decrease.

Food processing technologies can alter the composition of the constituents in raw materials, for example through structural changes or the degeneration of proteins, modifications of carbohydrates, shifts in lipid profiles and the loss of vitamins and minerals. Moreover, other properties, including physical characteristics such as the colour of a food product, might be changed [27]. In this context, fresh spinach leaves represent an interesting product because they are rich in vitamins, antioxidants, organic acids (with oxalate and malate as the most abundant) and alkaline mineral constituents [1,2]. Processing technologies might therefore have a considerable effect on the nutritional profile of the spinach. Since fresh spinach leaves are especially prone to microbial contamination, and thus sanitisation measures are indispensable to ensure food safety [3,4,5], it is essential to develop gentle disinfection methods that conserve the quality of the food. With regard to both aims, PAW has been found to be a very promising method [7,8,9], reducing bacterial loads and retaining polyphenols in food products [28]. Using a comprehensive metabolomic approach, we investigated the impact of PAW treatment on the quality and storability of baby spinach leaves in detail by analysing significant changes in the metabolite profiles of differently processed samples.

### 4.1. Storage Period as a Dominant Factor Determining Metabolite Profiles in Spinach Leaves

The metabolomics analysis produced extensive data of good quality, as demonstrated by the initial data processing steps for quality assurance. After normalisation, we were able to combine the datasets obtained by HILIC-HRMS and RPLC-HRMS, which generated a consolidated database for the subsequent statistical analyses. Our PCA model of the whole dataset clearly pointed at the storage period as the principal component causing the observable separation of the samples into two groups. In contrast, no significant changes were observable between C and TAP- or PAW-treated samples on D1 or D8 by PCA or OPLS-DA analysis. It appeared that the impact of the storage period was so strong that potentially existing small differences between the three treatment groups were completely overshadowed. We decided, therefore, that in the next step we would study the time-dependent changes separately for each treatment group and subsequently compare their respective trends to each other.

The OPLS-DA comparison of D1 versus D8 samples for each processing method showed significant differences for each pair. We were able to directly compare the multivariate analysis outputs of the pairwise differences because the non-differentiable D1 metabolomes of C, TAP and PAW samples could be considered as a common starting point. This way of proceeding not only promised greater accuracy, since it was based on the correlation coefficients of the annotated metabolites, but also appeared to be more appropriate than using ANOVA, which might deliver biased results due to the relatively low number of samples. We performed the metabolite comparisons separately for the outputs of HILIC-HRMS and RPLC-HRMS with the intention of detecting as many changed metabolic features as possible through this more refined workflow.

The metabolites that showed significantly changed intensities in the fresh spinach leaves during the 8-day storage period could be categorised into those sharing the same trend in C, TAP and PAW samples, those shared by two processing methods and those that were unique for one processing method. While the two shared-trend categories could be considered as representing general withering-related changes in the metabolite profiles of the stored spinach leaves, the uniquely occurring metabolites might be connected to the respective treatments. It should, however, be considered that we had applied a stringent cut-off for the identification of metabolites, so that some molecules designated as unique to one treatment or as shared by two treatments, such as citrate, histidine, adenine or cytidine, were excluded as non-significant in the others even though they were detected in all samples.

### 4.2. Characteristic Changes in the Metabolite Profiles of Stored Spinach Leaves

Quantitative changes in the metabolite profile during storage can either result from the release of water due to raw material shrinkage or from internal biochemical processes that alter molecule ratios. Water loss entails the leaching of hydrophilic compounds and an increase in more lipophilic constituents. However, this appears unlikely in the present experiment because the spinach leaves were air-packaged in sealed plastic bags before storage under refrigeration. This type of packaging appeared to have a positive impact also on preserving the chlorophyll, since typical degradation products such as pheophytins or chloroplast lipids were not found at significantly elevated levels at the end of the storage period [29,30,31].

Metabolites identified in all treatment groups with changing intensities from D1 to D8 included amino acids, carboxylic and sugar acids and nucleosides. These changes corresponded to some previously reported differences in metabolite occurrences that were associated with storage conditions and leaf senescence [32,33]. Senescence in plants is a transformational phase, where plant cells undergo complex changes in gene expression, metabolism and structure [11]. Anabolic physiological pathways are replaced mostly by catabolic reactions, disassembling macromolecules such as proteins, membrane lipids and RNA for recycling purposes and maintenance of function but ultimately leading to cell death. The transition can be induced by environmental factors such as temperature, daylight hours, drought, nutrient limitation, pathogen infection or by harvesting the green parts of the plant, leading to the browning and withering of green vegetables [34]. The transcriptomic analysis of gene expression changes in senescent processes showing an enhancement of the enzymes involved in protein and lipid degradation, ammonia and phosphorus mobilisation and antioxidant binding. Satisfactorily, the metabolite concentration changes that were shared by the differently processed spinach leaves in the present study corresponded well with the reported trends in early senescence. Thus, we considered that the elevated levels of free amino acids, carboxylic acids and sugar acids in all spinach-leaf samples after 8 days of refrigerated storage are mainly connected with adaption to the senescent state.

Spinach is 91.4% water and contains 2.9% proteins, peptides and free amino acids, 3.6% carbohydrates and 0.4% fats, of which mono- and poly-unsaturated fatty acids, e.g., linoleic acid, compose the main part [2]. Total protein hydrolysis revealed L-glutamate and L-aspartate as making up 23% of the total amino acid content [1], and unsurprisingly, we identified both amino acids in samples of all treatment groups, although at different levels. Furthermore, the ratio of aromatic amino acids was found to be high in spinach, and we detected significant levels of tyrosine, phenylalanine and tryptophan in the stored spinach leaves.

The release of amino acids through protease activities and the loss of amino acids from decarboxylation and deamination increases under withering. Intense proteolytic activity has been demonstrated in senescence-associated lytic vacuoles in different plant species, including cysteine proteases in spinach leaves [33]. Cysteine proteinase exists in several classes and plays multiple roles in plants. The discovery of a protease with an *N*-succinyl-Leu-Tyr-4-methylcoumaryl-7-amide cleavage activity in withering spinach leaves could help to explain the detection of increased succinate and tyrosine levels in our metabolomic analysis. Additionally, latent enzymes can become activated by changes in their physiological environment, e.g., by losing the stability of membrane associations, as was shown for phenolase during spinach leaf senescence [35]. Phenolase comprises the cresolase activity of tyrosinase, catalysing the ortho-hydroxylation of monophenols, which is an initial step in the metabolic pathway leading from L-tyrosine to L-hydroxyphenylalanine (L-DOPA) and subsequently melanin [36]. Accordingly, phenylalanine and derivatives were detected with significantly high intensities in all samples in the present study. Phenolase is, however, also involved in so-called enzymatic browning reactions, as quinone products may react with other metabolites, leading to decreased digestibility and nutritional value and changes of organoleptic properties in stored vegetables [12].

Under physiological conditions, the metabolism of the aromatic amino acids phenylalanine, tyrosine and tryptophan in plants is located upstream in pathways generating various growth hormones and secondary metabolites with multiple biological functions [37]. The three aromatic amino acids themselves are products of the shikimate pathway [37]. Phenylalanine is not only required for protein biosynthesis but also plays a role in cell survival by sustaining the vascularisation of the plant, counteracting oxidative stress [38]. The latter functions are of special relevance for cell maintenance in the early phase of senescence and could contribute to the upregulation of other mechanisms of cell protection. It was shown, e.g., that polyphenolic derivatives of aromatic amino acids help to preserve the nutritional quality of stored fruits by increasing injury-related organic acids, including malate and succinate, and antioxidants such as ascorbate [32,39,40,41]. Interestingly, we observed the prevalence of the same metabolites in the spinach leaves at D8. Tyrosine is a precursor of a multitude of secondary metabolites including vitamin E and phenols with antioxidant functions that can accumulate in leaf tissue [36,38,42]. Increased levels of tyrosine can therefore also be regarded as a countermeasure against cell degradation in the early stages of leaf senescence. The same might apply to tryptophan, which is at the starting point of a considerable number of metabolic pathways leading to the production of the plant hormone auxin, tryptamine derivatives including serotonin, phytoalexins and antioxidants such as melatonin, indole glucosinolates and terpenoid indole alkaloids. Accordingly, tryptophan plays a pivotal role in the regulation of plant growth and development and in stress responses [37,43]. A number of studies examining senescence-related functions of tryptophan in detached leaves have reported increased biosynthesis of this amino acid as well as of major derived metabolites in a regulated process [44,45].

L-glutamate and L-aspartate levels in the spinach leaves decreased during the storage period, which was probably a consequence of the metabolic transition from nitrogen assimilation to remobilisation in plant senescence [46]. It has been found that enzymes involved in the primary assimilation of nitrogen, such as glutamate synthase, are less expressed in this phase, whereas glutamate dehydrogenase expression is upregulated, producing α-ketoglutarate and ammonia. Moreover, photorespiratory nitrogen transamination by glutamate:glyoxylate aminotransferase could lead to a reduction in glutamate concentrations [47]. The content of free aspartate in the spinach leaves decreased during storage, probably because this amino acid acts similarly to glutamate as a nitrogen carrier and contributes to its transport, recycling and storage [48]. Aspartate is also involved in the biosynthesis of other amino acids such as asparagine, lysine, methionine, threonine and isoleucine, and is a building block in the production of nucleic acids. Moreover, aspartate deamination leads to oxaloacetate, which can be channelled into the tricarboxylic acid (TCA) cycle in mitochondrial respiration.

The involvement of the TCA cycle in the developing senescence in the stored spinach leaves was evident as the concentrations of several metabolites essential for the TCA cycle were modified. Since we observed a decrease in malate, citrate and cis-aconitate concentrations, it appeared that the detached leaves tried to sustain the production of energy and electron donors by draining the amino acid reservoir through transaminase activities and by using the available sugar acids. In this regard, the detected slight increase of succinate in the spinach leaves may present a transitional state resulting, e.g., from the inhibition of succinate dehydrogenase by malonate and/or the influx of fresh precursors derived from glutamate as α-ketoglutarate by glutamate dehydrogenase activity or via the γ-aminobutyric (GABA) shunt by glutamate decarboxylase activity [49].

The storage-related changes in the amino acid and sugar acid metabolism in the spinach leaves also affected the interconnected pathways of nucleotides and lipids. The levels of purine and pyrimidine nucleotides, which are synthesised de novo or by salvage pathways from amino acid and sugar precursors [50], were decreased at D8. It has been reported that gene expression is generally downregulated in leaf senescence and that RNA synthesis, as well as the synthesis of RNA nucleotides, is mostly halted [10,11,34]. Furthermore, the decline in RNA levels is accelerated by increased RNAse activities. The senescent state in green leaves is also correlated with increased lipid oxidation and membrane permeability and fluidity, resulting from lipid depletion through upregulated phospholipase D activity and lower availability of amino-acid- and sugar-acid-derived acetyl-CoA building blocks for lipid synthesis [30,51,52]. Accordingly, we were able to observe the decline of free fatty acids in the spinach after the storage period. Senescent effects compromising membrane lipids were also reflected in our detection of decreasing choline levels in the spinach leaves. Choline can be transformed into different phospholipids but also into glycine betaine aldehyde, a precursor of the osmoregulator trimethylglycine [53]. This choline derivative, which is rather abundant in spinach, has relevant antioxidant activity and helps to preserve the membrane integrity against oxidative stress. It was shown that treatment with choline chloride could lessen the browning of fruits, supposedly by replenishing their glycine betaine pool [54]. We assumed therefore that the decreasing choline concentrations in the stored spinach leaves were connected to the mitigation of senescent effects.

Plant hormones have an important role in regulating plant senescence [10]. Among the metabolites with declining levels in the spinach samples at D8 we discovered several with relevance for hormone activity, such as pyridoxal and benzoic acid derivatives. Pyridoxal-5′-phosphate is a cofactor of 1-aminocyclopropane-1-carboxylic acid (ACC) synthase, the key enzyme in the pathway leading from methionine to ethylene, which is essential for fruit ripening but also for the regulation of plant growth and development. Post-harvest, ethylene accelerates leaf senescence but is counteracted by polyamines such as putrescine and spermidine that are derived from arginine and ornithine [55]. Interestingly, spermidine levels were likewise decreased in all spinach samples, indicating a controlled transition to the senescent state. Benzoic-acid-related derivatives include the plant defence hormone salicylic acid, which is synthesised from phenylalanine via benzoic acid as an intermediate [56]. The progression of senescence appears to be correlated with salicylic acid levels in leaves, and increased conversion of the hormone to 2,3-dihydroxybenzoic acid by salicylic acid 3-hydroxylase is considered to be a regulation mechanism.

Taking these findings together, the broad impact of the onset of senescence on multiple metabolic pathways was clearly traceable in the metabolome of the stored spinach leaves. Most changes occurred independently of the applied processing methods, showing that TAP or PAW had very little influence on internal metabolic mechanisms during the transition to the senescent state. It appeared, therefore, that the PAW treatment of fresh spinach leaves has a positive effect regarding aspects of food safety [9] but that it is less efficient in conserving nutrient contents and thus the food quality. However, we also detected a few metabolites that changed uniquely with the respective treatments. They provided grounds for a more detailed evaluation of PAW in terms of food quality and shelf-life extension.

### 4.3. PAW Treatment Sustains Anti-Senescent Mechanisms in Detached Spinach Leaves

Unique changes occurring during the storage period in the differently treated spinach leaves were discovered by comparison of the respective metabolite profile trends. Since the annotation of some metabolites was partly impaired by equivocal MS/MS spectra, the selection of metabolites as characteristic for one group was probably somewhat biased. Furthermore, our stringent filtering for statistical significance was shown to have some misguiding side effects, such as, e.g., one metabolite, norleucine, identified as specific for changes in the control samples was also determined in the two other groups, although with non-significant trends. Among the metabolites found for the TAP samples, two could not be clearly identified, whereas the annotated glycine betaine is strongly connected to the changes in the choline levels that were detected for all treatment groups. Furthermore, the determination of creatinine should be considered with some reservations, since this amine occurs mainly in vertebrates as a breakdown product of muscle creatine and has been measured in vegetable materials only in trace amounts [57]. Thus, there were, in fact, no metabolites unambiguously recognisable for storage-induced changes in C and TAP samples.

In PAW samples, however, several metabolites were discernible as uniquely changed, which could be considered plausible with greater confidence. Whereas a significant increase was detected for a flavonoid-7-O-glucuronide derivative in PAW-treated samples, it was decreased in control samples, showing an inverse trend. Flavonoids represent the major class of plant polyphenols with a relevant antioxidant capacity [55]. Their intake is considered beneficial for human health. Consequently, the flavonoid level in vegetables is a decisive factor for the evaluation of food quality. Spinach has a substantial total flavonoid content of about 1000 mg/kg green mass, which normally remains relatively stable, although individual derivatives can vary considerably depending on the environmental conditions in the growth period and the post-harvest storage conditions [58,59]. Important flavonoid derivatives in spinach include spinacetins, patuletins, jaceidins, glucuronides and acylated di- and triglycosides of methylated and methylenedioxy derivatives of 6-oxygenated flavonols [60]. The flavonoid-7-O-glucuronide that we were able to annotate in the spinach leaves processed with PAW was identified as methylenedioxyflavone-glucuronide (5,3′,4′-trihydroxy-3-methoxy-6:7-methylene-dioxyflavone-4′-βD-glucuronide), which has been described as a major flavonoid constituent in baby spinach leaves [58]. On average, it makes up 43% of the total flavonoid content, so that protection of this metabolite by PAW treatment, as indicated by our results, is of considerable consequence for maintaining the health-promoting antioxidant capacity of the spinach leaves.

Remarkably, we could detect a second plant polyphenol in the group of uniquely changed metabolites in the PAW group. Cinnamic acid derivatives such as hydroxycinnamic acids are phenolic acids with antioxidant, antimicrobial and anti-inflammatory activities. An important subgroup includes coumaric acid derivatives, of which we additionally determined 2-(p-coumaroyl)malate as the major metabolite in a cluster that also contained trans-4-coumaric acid and coumaroylquinic acid, which was increased in PAW samples and decreased in C and TAP samples. In contrast, we found a decline in 1-O-feruloylglucose, a hydroxycinnamic acid acylglycoside present in various vegetables and fruits and in all spinach samples, independently of the treatment. The compound acts as an acyl donor in pathways forming ferulic acid conjugates such as betacyanin pigments and feruloyltyramine, a metabolite involved in the plant’s defence response to external stimuli [61]. Upregulation of the catalysing enzyme tyramine hydroxycinnamoyl transferase has been observed in stressed plants requiring sufficient substrate contents. Accordingly, we also detected increased tyramine and methoxytyramine levels in the PAW-treated samples and significantly decreased levels in C and TAP samples. Taken together, our findings on polyphenols clearly suggested that mechanisms of stress resistance had been intensified in the spinach leaves by the application of PAW processing.

The inverse trends observed in concentrations of several amino acids such as glutamine, valine and arginine indicated a certain impact of PAW treatment on the extent of leaf nitrogen remobilisation in the early-stage senescence of the spinach leaves [46]. Furthermore, PAW might also influence the composition of the lipid acid pool during the senescent transition, to a certain degree. We determined a significantly lower level of γ-linolenic acid in the PAW-treated samples compared to the C and TAP samples. It has previously been reported that triacylglycerols in senescing leaves are enriched in γ-linolenic acid due to the transfer of fatty acids from de-esterified chloroplast galactolipids to cytosolic triacylglycerols [62]. Free γ-linolenic acid is also a substrate to lipoxygenase and a precursor of ethylene, malondialdehyde and the plant growth regulator jasmonic acid, which are all involved in promoting senescence. Therefore, the PAW processing may possibly have helped to slow down the senescence development in the detached spinach leaves.

### 4.4. Enhancement of Carbohydrate Pathways by PAW Treatment May Have a Protective Effect

Our considerations based on the occurrences of individual metabolites were further supported by results from the metabolic pathway enrichment analysis. Our observation that most of the pathways specifically affected by the PAW treatment were connected with carbohydrate metabolism indicated a significant effect on senescence-related metabolic activities in the spinach leaves. It has been shown that the loss of sugars contributes to accelerated post-harvest senescence in green vegetables, leading subsequently to a decline in other nutrients such as antioxidants [63]. When the impact of different processing methods on the free sugar content in broccoli was analysed, it could be demonstrated that treatments causing a slight stress in the plant material delayed the onset of senescence because of countermeasures in the recovering tissue, which included the upregulation of carbohydrate pathways. It is thus plausible to assume that the stimulus provided by PAW processing had a similar effect on the spinach leaves, which resulted in a certain degree of protection in early-stage senescence.

## 5. Conclusions

The metabolomic analysis of differently processed spinach leaves after 8 days of refrigerated storage showed metabolome changes consistent with the transition into the senescent state. We did not detect significant differences between control, TAP- and PAW-treated samples, either immediately after processing or after 8 days of storage. However, when comparing the progression of the changes, we detected differences with regard to the occurrence of specific polyphenolic antioxidants, some amino acids and γ-linolenic acid in PAW-treated spinach leaves, which are all connected with plant defence mechanisms against stress. The subsequent metabolic pathway analysis revealed the enhancement of carbohydrate-related reactions, which also is connected to a delay in senescence development. It thus appeared that PAW processing, using the setup described here, could possibly facilitate the preservation of valuable nutrients, thereby contributing to the food quality and shelf-life of the spinach leaves.

## Figures and Tables

**Figure 1 foods-10-03067-f001:**
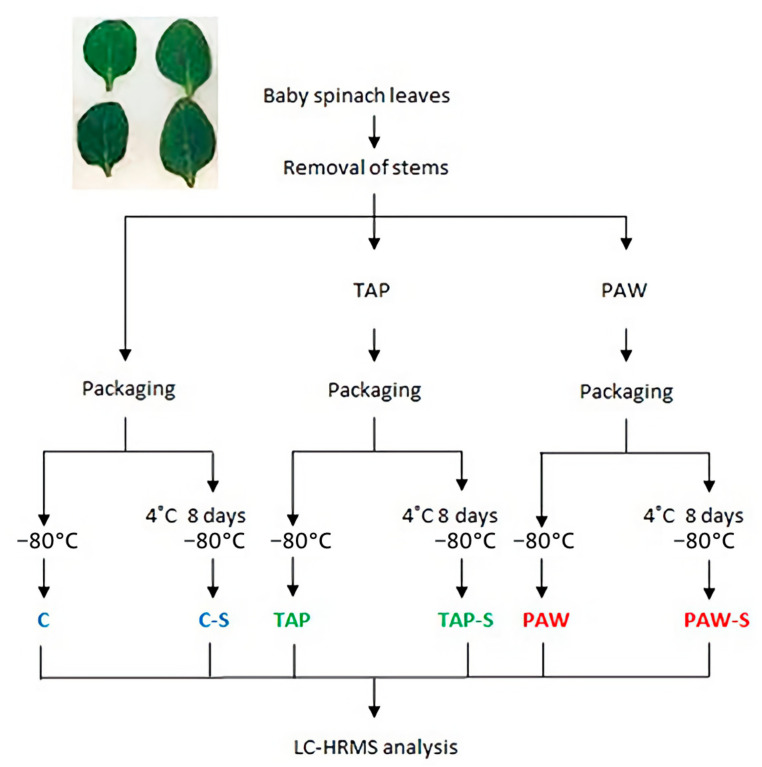
Processing of fresh baby spinach leaves resulting in six treatment groups with *n* = 3 replicates per group. C: untreated control; C-S: untreated control stored for 8 days; TAP: tap-water-rinsed; TAP-S: tap-water-rinsed and stored for 8 days; PAW: plasma-activated-water-processed; PAW-S: plasma-activated-water-processed and stored for 8 days.

**Figure 2 foods-10-03067-f002:**
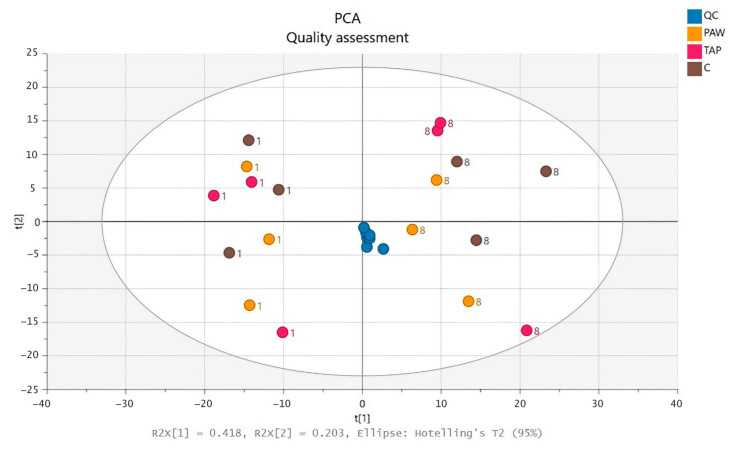
PCA scores plot including QCs and all samples from LC-HRMS analysis. Quality control (QC); untreated control spinach (C: 1; C-S: 8); tap-water-rinsed spinach (TAP: 1; TAP-S: 8); plasma-activated-water-treated spinach (PAW: 1; PAW-S: 8).

**Figure 3 foods-10-03067-f003:**
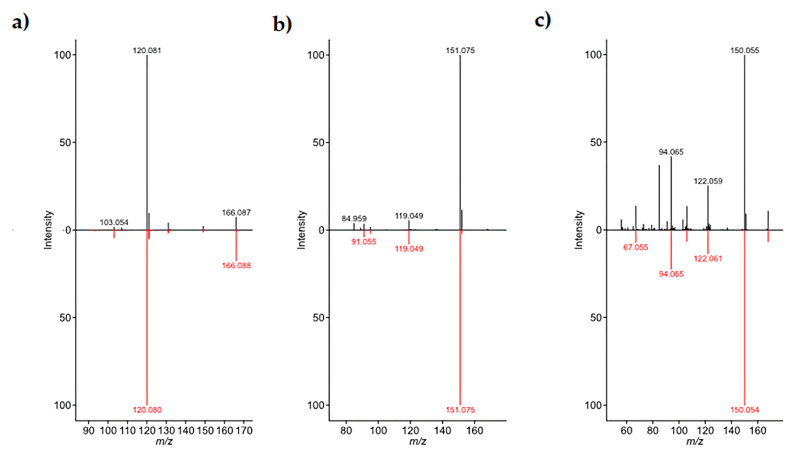
Measured MS/MS spectra (up, in black) matched to the respective reference spectra from the MS-DIAL curated library (down, in red) for (**a**) phenylalanine (MS/MS of [M − H]^−^), (**b**) pyridoxal (MS/MS of [M + H]^+^) and (**c**) methoxytyramine (MS/MS of [M + H]^+^).

**Figure 4 foods-10-03067-f004:**
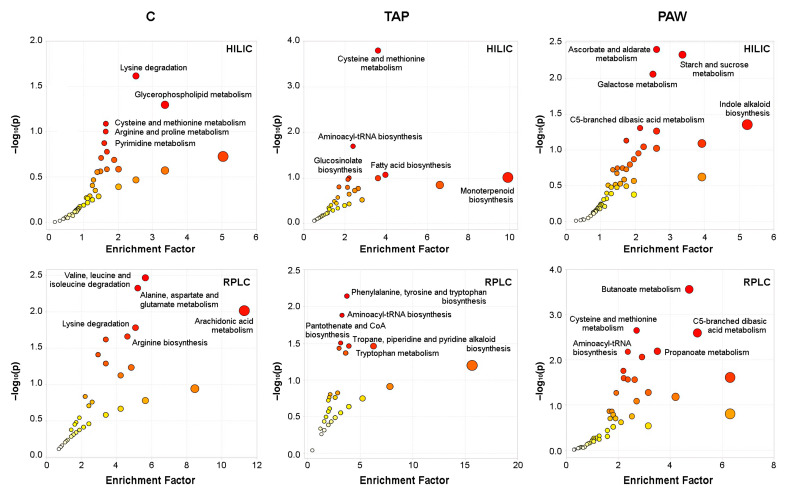
KEGG pathway association of metabolites changed during storage (D1 vs. D8) in differently processed spinach leaves (C, TAP or PAW) and identified by HILIC- or RPLC-HRMS. The MS Peaks to Pathway analysis by *mummichog* shows significantly affected pathways (see Appendix A) as red dots, whose sizes are dependent on their connected *p*-values (see Section 2.7.2). The five most-changed pathways are named for each sample group.

**Table 1 foods-10-03067-t001:** Concentration of nitrogen species, hydrogen peroxide, pH and ORP in PAW and TAP used in the spinach processing experiment.

	NO_2_ (mg/L)	NO_3_ (mg/L)	H_2_O_2_ (mg/L)	pH	ORP (mV)
TAP	ND	ND	ND	8.0 ± 0.1	−45.3 ± 2.4
PAW	32.4 ± 5.6	462.3 ± 1.2	8.8 ± 0.4	2.3 ± 0.1	284.1 ± 11.5

ND: Non detected.

**Table 2 foods-10-03067-t002:** Summary of multivariate models for quality assessment by PCA and differently processed spinach samples by OPLS-DA after analysis by LC-HRMS.

Type	Model	Time Points	LV	R^2^X	R^2^Y	Q^2^	*p*-Value
PCA	all samples	all	3	0.691		0.405	
OPLS-DA	C vs. TAP vs. PAW	D1	1 + 0 + 0	0.257	0.412	−0.0244	NS
OPLS-DA	C-S vs. TAP-S vs. PAW-S	D8	1 + 0 + 0	0.210	0.376	−0.0176	NS
OPLS-DA	C vs. C-S	D1 vs. D8	1 + 0 + 0	0.618	0.961	0.933	**0.017**
OPLS-DA	TAP vs. TAP-S	D1 vs. D8	1 + 0 + 0	0.764	0.993	0.967	**0.049**
OPLS-DA	PAW vs. PAW-S	D1 vs. D8	1 + 0 + 0	0.790	0.993	0.971	**0.042**

PCA: principal component analysis; OPLS-DA: orthogonal partial least-squares discriminant analysis; LV: latent variable; R^2^X: total explained variance; R^2^Y: goodness of fit; Q^2^: predictive ability; *p*-value: ≤ 0.05 considered as significant (bold) (CV-ANOVA); NS: not significant.

**Table 3 foods-10-03067-t003:** Shared and unique relevant metabolites extracted from the comparison of C, TAP and PAW samples before (D1) and after 8 days of storage at 4 °C (D8).

LC Mode	Ionisation Mode	Average RT	Average *m*/*z*	Cluster Size ^a^	Ion	Neutral Formula	Tentative Annotation (Spectral Match)	CANOPUS ^b^ Level	ClassyFire	Ann. Level ^c^	D1 vs. D8					
											C *p*(corr) ^d^	C *p*( )	TAP *p*(corr) ^d^	TAP *p*( )	PAW *p*(corr) ^d^	PAW *p*( )
**Shared by All Processing Methods**																
RPLC	pos	1.97	166.0857	6	[M + H]^+^	C_9_H_11_NO_2_	Phenylalanine	7	Phenylalanine and deriv.	2	1.00	0.48	0.84	0.44	0.97	0.43
HILIC	neg	9.86	164.0714	3	[M − H]^−^	C_9_H_11_NO_2_	Phenylalanine	7	Phenylalanine and deriv.	2	0.98	0.08	0.94	0.07	0.90	0.06
RPLC	pos	2.35	205.0968	16	[M + H]^+^	C_11_H_12_N_2_O_2_	Tryptophan	7	α-amino acids	2	0.98	0.24	0.92	0.27	0.97	0.27
HILIC	neg	12.27	175.0608	2	[M − H]^−^	C_7_H_12_O_5_	2-Isopropylmalate	6	Meth. branched fatty acids	2	0.99	0.21	0.84	0.20	0.92	0.21
RPLC	neg	2.66	175.0608	1	[M − H]^−^	C_7_H_12_O_5_	2-Isopropylmalate	6	Meth. branched fatty acids	2	0.99	0.12	0.66	0.11	0.81	0.13
RPLC	pos	1.20	182.0808	5	[M + H]^+^	C_9_H_11_NO_3_	Tyrosine	7	Tyrosine and deriv.	2	0.78	0.16	0.83	0.19	0.78	0.15
HILIC	pos	12.99	182.0808	3	[M + H]^+^	C_9_H_11_NO_3_	Tyrosine	7	Tyrosine and deriv.	2	0.95	0.13	0.95	0.12	0.74	0.09
HILIC	neg	15.64	117.0192	2	[M − H]^−^	C_4_H_6_O_4_	Succinate	Subclass	Dicarbox. acids and deriv.	2	0.90	0.14	0.92	0.17	0.94	0.19
HILIC	pos	14.49	120.0652	2	[M + H]^+^	C_4_H_9_NO_3_	L-(−)-Threonine	7	α-amino acids	2	0.96	0.11	0.95	0.10	0.91	0.09
HILIC	pos	15.43	133.0605	1	[M + H]^+^	C_4_H_8_N_2_O_3_	Asparagine	7	α-amino acids	2	0.91	0.08	0.98	0.09	0.74	0.08
HILIC	neg	12.94	135.0305	2	[M − H]^−^	C_4_H_8_O_5_	Threonate	Subclass	β-hydroxy acids and deriv.	2	0.94	0.08	0.96	0.10	0.94	0.10
HILIC	pos	3.88	679.2970	5	[M + H]^+^	C_35_H_42_N_4_O_10_	Unknown	6	Oligopeptides	3	0.89	0.08	0.77	0.08	0.74	−0.05
RPLC	pos	1.14	245.0764	4	[M + H]^+^	C_9_H_12_N_2_O_6_	Uridine	Class	Pyrimidine nucleosides	2	−0.83	−0.05	−0.89	−0.07	−0.89	−0.08
RPLC	pos	2.80	374.1438	24	[M + NH_4_]^+^	C_16_H_20_O_9_	1-O-Feruloylglucose	5	Coumaric acids and deriv.	2	−0.98	−0.12	−0.92	−0.10	−0.69	−0.05
HILIC	pos	14.73	148.0603	4	[M + H]^+^	C_5_H_9_NO_4_	L-Glutamate	7	Glutamic acid and deriv.	2	−0.67	−0.09	−0.85	−0.12	−0.53	−0.09
RPLC	pos	0.67	455.1137	1	[M + Na]^+^	C_23_H_20_N_4_O_3_S	Asperulosidic acid	6	Phenolic glycosides	3	−0.91	−0.06	−0.88	−0.08	−0.99	−0.12
RPLC	pos	0.96	168.0652	1	[M + H]^+^	C_8_H_9_NO_3_	Pyridoxal	6	Aryl-aldehydes	2	−0.92	−0.08	−0.90	−0.09	−0.96	−0.10
HILIC	pos	7.39	168.0652	1	[M + H]^+^	C_8_H_9_NO_3_	Pyridoxal	Subclass	Pyridine carboxaldehydes	2	−0.92	−0.05	−0.92	−0.06	−0.90	−0.07
HILIC	neg	15.11	132.0300	1	[M − H]^−^	C_4_H_7_NO_4_	L-Aspartate	8	L-α-amino acids	2	−0.89	−0.08	−0.99	−0.09	−0.73	−0.06
RPLC	pos	1.26	307.0830	1	[M + H]^+^	Unknown	Unknown	Subclass	1-hydoxy-2-unsubstituted benzenoids	4	−0.89	−0.08	−0.97	−0.09	−0.87	−0.09
RPLC	neg	0.71	133.0138	4	[M − H]^−^	C_4_H_6_O_5_	D-(+)-Malate	Subclass	β-hydroxy acids and deriv.	2	−0.90	−0.14	−0.55	−0.07	−0.64	−0.06
RPLC	pos	0.55	146.1649	3	[M + H]^+^	C_7_H_19_N_3_	Spermidine	6	Dialkylamines	2	−0.82	−0.15	−0.92	−0.15	−0.66	−0.09
HILIC	pos	21.60	104.1067	2	[M + H]^+^	C_5_H_14_NO	Choline	5	Tetraalkylammonium salts	2	−0.62	−0.23	−0.73	−0.18	−0.97	−0.24
RPLC	pos	1.56	268.1030	10	[M + H]^+^	C_10_H_13_N_5_O_4_	Adenosine	Class	Purine nucleosides	2	−0.96	−0.23	−0.96	−0.26	−0.93	−0.32
**Shared by Two Processing Methods**																
**TAP and PAW**																
RPLC	pos	1.13	101.0232	4	[M + H]^+^	C_4_H_4_O_3_	Succinic anhydride	Subclass	Dicarbox. acids and deriv.	2	0.86	0.04	0.85	0.05	0.94	0.06
HILIC	pos	10.96	132.1016	4	[M + H]^+^	C_6_H_13_NO_2_	DL-Norleucine	6	β-amino acids and deriv.	3	0.31	0.04	0.59	0.10	0.54	0.06
RPLC	pos	1.89	199.1913	1	[M + H]^+^	C_10_H_22_N_4_	Unknown	Subclass	Guanidines	4	−0.11	−0.01	−0.97	−0.20	−0.78	−0.19
HILIC	pos	11.98	130.0861	1	[M + H]^+^	C_6_H_11_NO_2_	Unknown	6	Amino acids	3	0.39	0.03	−0.58	−0.05	−0.62	−0.11
HILIC	pos	25.21	123.0550	1	[M + H]^+^	C_6_H_6_N_2_O	Nicotinamide	6	Nicotinamides	2	−0.61	−0.03	−0.81	−0.05	−0.79	−0.06
RPLC	pos	5.19	200.2009	1	[M + H]^+^	C_12_H_25_NO	Unknown	Subclass	Fatty amides	3	−0.92	−0.04	−0.95	−0.05	−0.98	−0.05
**C and TAP**																
RPLC	pos	3.70	679.2974	3	[M + H]^+^	C_35_H_42_N_4_O_10_	Unknown	6	α-amino acids and deriv.	3	0.73	0.06	0.62	0.07	0.40	0.02
RPLC	pos	4.53	615.2780	5	[M + H]^+^	-	Unknown	Subclass	Billirubins	4	0.87	0.05	0.70	0.05	0.89	0.04
HILIC	pos	10.38	132.1016	3	[M + H]^+^	C_6_H_13_NO_2_	DL-Norleucine	7	α-amino acids	3	0.59	0.09	0.34	0.08	−0.21	−0.04
RPLC	pos	5.46	368.4239	1	[M + H]^+^	Unknown	-	-	-	4	−0.92	−0.06	−0.70	−0.05	0.04	0.00
HILIC	neg	16.60	133.0139	6	[M − H]^−^	C_4_H_6_O_5_	D-(+)-Malate	Subclass	β-hydroxy acids and deriv.	3	−0.56	−0.11	−0.72	−0.13	−0.36	−0.06
RPLC	pos	5.14	279.1588	19	[M + H]^+^	C_16_H_22_O_4_	Unknown	5	Benzoic acids	4	−0.84	−0.20	−0.70	−0.17	−0.34	−0.08
RPLC	pos	0.70	360.1485	1	[M + NH_4_]^+^	C_12_H_22_O_1_1	-	5	Disaccharides	3	−0.70	−0.05	−0.82	−0.05	−0.81	−0.04
**C and PAW**																
HILIC	pos	4.86	242.1283	1	[M + H]^+^	C_14_H_15_N_3_O	Unknown	Subclass	Cinnamic acid amides	3	0.57	0.05	0.42	0.02	0.75	0.08
HILIC	pos	18.01	156.0765	1	[M + H]^+^	C_6_H_9_N_3_O_2_	Histidine	7	Histidine and deriv.	3	0.87	0.06	0.68	0.04	0.86	0.07
HILIC	neg	18.59	191.0193	2	[M − H]^−^	C_6_H_8_O_7_	Citrate	Subclass	Tricarboxylic acids and deriv.	3	−0.90	−0.14	−0.35	−0.04	−0.50	−0.08
HILIC	neg	18.10	173.0091	1	[M − H]^−^	C_6_H_6_O_6_	Aconitate	Subclass	Tricarboxylic acids and deriv.	3	−0.65	−0.05	−0.62	−0.04	−0.63	−0.05
**Unique for One Processing Method**																
**C**																
RPLC	pos	1.48	132.1016	4	[M + H]^+^	C_6_H_13_NO_2_	DL-Norleucine	7	α-amino acids	3	−0.82	−0.29	−0.19	−0.09	−0.61	−0.21
**TAP**																
HILIC	pos	9.26	114.0660	1	[M + H]^+^	C_4_H_7_N_3_O	Creatinine	6	α-amino acids and deriv.	2	0.45	0.00	−0.51	−0.05	−0.40	−0.01
RPLC	pos	2.52	155.1541	1	[M + H]^+^	Unknown	-	-	-	4	−0.16	0.00	−0.69	−0.18	−0.30	0.00
RPLC	pos	3.78	171.1489	1	[M + H]^+^	C_9_H_18_N_2_O	-	Class	Organic carbonic acids and deriv.	3	−0.80	−0.01	−0.61	−0.14	−0.67	−0.01
HILIC	pos	10.76	118.0859	1	[M + H]^+^	C_5_H_11_NO_2_	Glycine Betaine	-	-	4	−0.43	−0.08	−0.67	−0.17	0.05	0.02
**PAW**																
RPLC	pos	3.12	242.1284	1	[M + H]^+^	C_14_H_15_N_3_O	-	Subclass	Cinnamic acid amides	3	0.40	0.05	0.02	0.00	0.67	0.11
HILIC	neg	14.20	165.0192	1	[M − H − H_2_O]^−^	C_8_H_8_O_5_	-	Subclass	α-hydroxy acids and deriv.	3	0.87	0.04	0.01	0.00	0.98	0.06
RPLC	neg	5.33	447.2512	1	[M − H]^−^	Unknown	-	-	-	4	−0.48	−0.01	−0.71	−0.03	−0.89	−0.06
HILIC	neg	3.46	277.2166	3	[M − H]^−^	C_18_H_30_O_2_	γ-Linolenic acid	Subclass	Glycerophosphocholine	3	0.04	0.00	0.30	0.04	−0.88	−0.10
HILIC	pos	11.60	244.0924	1	[M + H]^+^	C_9_H_13_N_3_O_5_	Cytidine	Class	Pyrimidine nucleosides	2	−0.73	−0.03	−0.93	−0.04	−0.85	−0.05
HILIC	pos	8.98	136.0616	1	[M + H]^+^	C_5_H_5_N_5_	Adenine	5	6-aminopurines	2	−0.70	−0.03	−0.66	−0.04	−0.58	−0.05
RPLC	pos	5.62	496.3395	2	[M + H]^+^	C_24_H_50_NO_7_P	Lysophosphatidylcholine (16:0)	5	Lysophosphatidylcholines	2	−0.55	−0.04	−0.02	0.00	−0.77	−0.07
HILIC	neg	4.68	519.0775	2	[M − H]^−^	C_23_H_20_O_14_	5,3’,4’,-Trihydroxy-3-methoxy-6:7-methylenedioxyflavone 4’-βD-glucuronide	7	Flavonoid-7-O-glucuronides	3	0.42	0.03	−0.15	−0.01	−0.63	−0.07
**Inverse**																
RPLC	pos	1.79	168.1014	3	[M + H]^+^	C_9_H_13_NO_2_	Methoxytyramine	Subclass	Phenethyl amines	2	−0.86	−0.08	−0.86	−0.07	0.71	0.07
HILIC	pos	15.03	147.0763	3	[M + H]^+^	C_5_H_10_N2O_3_	Glutamine	7	α-amino acids	2	−0.55	−0.09	0.19	0.05	0.37	0.06
HILIC	pos	12.22	118.0859	1	[M + H]^+^	C_5_H_11_NO_2_	Valine	7	α-amino acids	2	0.50	0.05	0.19	0.03	−0.78	−0.07
RPLC	pos	0.63	175.1184	1	[M + H]^+^	C_6_H_14_N_4_O_2_	Arginine	8	L-α-amino acids	2	0.78	0.11	0.32	0.04	−0.50	−0.07
RPLC	pos	5.71	282.2782	1	[M + H]^+^	C_18_H_35_NO	-	Subclass	Fatty amides	2	−0.42	−0.03	0.45	0.07	−0.61	−0.05
RPLC	neg	3.12	279.0504	12	[M − H]^−^	C_13_H_12_O_7_	2-(p-coumaroyl)malate	6	Coumaric acid esters	3	−0.85	−0.07	−0.74	−0.05	0.58	0.04
RPLC	pos	1.38	138.0913	2	[M + H]^+^	C_8_H_11_NO	Tyramine	5	Aralkylamines	2	−0.95	−0.07	−0.69	−0.05	0.66	0.03

The metabolites in each category are sorted according to probability (sum of *p*(corr) of C, TAP and PAW). RPLC: reverse-phase liquid chromatography; HILIC: hydrophilic interaction liquid chromatography; pos: positive; neg: negative; RT: retention time. The MS/MS spectra of underlined metabolites are also shown in Figure 3. ^a^ Cluster size: clusters can contain differently charged ion adducts of a metabolite (Appendix A). The metabolic feature with the most intense ion is shown in Table 3. ^b^ CANOPUS level: CANOPUS provided the possibility of assigning compound classes to otherwise unidentified features for which no spectral reference data were available. The classification considers the structure-based chemical taxonomy (ChemOnt) built using ClassyFire that uses only chemical structures and structural features for the automatic assignment of all known chemical compounds to a taxonomy consisting of >4800 different categories. ChemOnt is organised as a tree, where the Kingdom is either Organic compounds or Inorganic compounds. Superclasses such as Lipids and Lipid-like Molecules, and Benzenoids are categories in a Kingdom. Pyrimidine nucleosides constitute a class, whereas guanidine is an example of a subclass. There can be up to 11 levels in the ontology. ^c^ Annotation level of identification: level 1—unambiguously identified metabolite; level 2—putatively identified metabolite; level 3—tentatively characterised metabolite class; level 4—unidentified or unclassified metabolite, but differentiable. ^d^ The direction of the change is indicated by the plus/minus sign in front of the number.

**Table 4 foods-10-03067-t004:** Comparison of KEGG^a^ pathways allocated to metabolite changes observed during storage (D1 to D8) in differently processed spinach leaves.

C	TAP	PAW
Alanine, aspartate and glutamate metabolism		Alanine, aspartate and glutamate metabolism
	Aminoacyl-tRNA biosynthesis	Aminoacyl-tRNA biosynthesis
		Ascorbate and aldarate metabolism
		Butanoate metabolism
		C5-branched dibasic acid metabolism
		Cyanoamino acid metabolism
	Cysteine and methionine metabolism	Cysteine and methionine metabolism
		Galactose metabolism
	Phenylalanine, tyrosine and tryptophan biosynthesis	
		Propanoate metabolism
		Starch and sucrose metabolism
Valine, leucine and isoleucine degradation		

^a^ KEGG pathways, s. Appendix A. Uniquely changed metabolite pathways are underlined in the respective treatment groups.

## Data Availability

The raw data generated is available upon request.

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
