# Peer review of "Impact of Plasma-Activated Water Treatment on Quality and Shelf-Life of Fresh Spinach Leaves Evaluated by Comprehensive Metabolomic Analysis"

_foods, 2021, doi:10.3390/foods10123067_

Round 1

Reviewer 1 Report

The study entitled "Impact of plasma-activated water treatment on quality and shelf-life of fresh spinach leaves evaluated by comprehensive metabolomic analysis" used comprehensive metabolomics analysis to evaluate the possible changes in the metabolite content of spinach leaves treated by plasma activated water treatment. The results showed that paw treatment significantly improved the stress resistance and healthy antioxidant capacity of spinach samples, and delayed the aging development of spinach samples. The manuscript has some positive elements, but it lacks novelty. Experiments need to be added and substantive changes need to be made.

The author should consider the following points:

——Main points:

1.There are many simple mistakes in the manuscript. If the authors could have read it one more time, it would be easy for them to correct those mistakes.

2.The manuscript is not concise, so it should be extensively revised. Especially in the Results and Discussion section, please include the new ideas in a short form. Try to reduce the information and condense it as much as possible.

3.Only describing a large amount of data is fairly preliminary and does not provide novel insights. The authors are recommended to follow up the literature about PAW treatment to find more interesting regulatory metabolites and for some major concern substances, quantitative or other tests should be added to further explain.

--Minor points:

1.Please check the use of ‘a/an’. Such as at Ln14, 58, 118, 209, 222, 317, 361 548 and 555.

2.Ln11, please change “which is a result of leaf senescence”.

3.Ln50-51。 The position of this sentence seems inappropriate.

4.Ln78。" sregular"? Please revise this sentence.

5.Ln94。Oxygen? Oyxgen?

6.Ln226。 Please add "were" after the sample. sotrage?

7.Ln260-261。 Please revise this sentence.

8.Ln287。 Is it relevant?

9.Ln293。 Statistically significant.

10.Please change the position of the legend in Figure 3.

11.Ln454。 Please add 'is' after and.

12.Ln537。 Please delete "that".

Author Response

We would like to thank you for the interest shown in our article and appreciate the feedback provided. We have made the appropriate changes following the recommendations of the reviewers.

Here we present a detailed response to major and minor comments from the reviewers:

REVIEWER 1

The study entitled "Impact of plasma-activated water treatment on quality and shelf-life of fresh spinach leaves evaluated by comprehensive metabolomic analysis" used comprehensive metabolomics analysis to evaluate the possible changes in the metabolite content of spinach leaves treated by plasma-activated water treatment. The results showed that paw treatment significantly improved the stress resistance and healthy antioxidant capacity of spinach samples, and delayed the ageing development of spinach samples. The manuscript has some positive elements, but it lacks novelty. Experiments need to be added and substantive changes need to be made.

Response: Thank you for reviewing the manuscript. As recognised, it describes the assessment of a possible impact of PAW treatment on the metabolite composition of fresh spinach leaves. While PAW has been applied to spinach before, evaluating effects on the bacterial load, changes in the metabolite profiles in the leaves have never been studied before. Moreover, we used an extensive chemometrics approach for the identification of significant changes, which adds further novelty to the study. The results of the experiments performed were conclusive and clearly showed that metabolites connected to senescence development were affected by the PAW treatment. While additional experiments, e.g. with a prolonged storage period, could be interesting to perform in a follow-up study, we consider the current findings as consistent and as clearly indicating the benefit of PAW for increasing the shelf life of spinach leaves.     

The author should consider the following points:

Main points:

  1. There are many simple mistakes in the manuscript. If the authors could have read it one more time, it would be easy for them to correct those mistakes.

Response: We have revised the manuscript thoroughly. Thank you for your detailed suggestions for improvements as given in the Minor points.

  1. The manuscript is not concise, so it should be extensively revised. Especially in the Results and Discussion section, please include the new ideas in a short form. Try to reduce the information and condense it as much as possible.

Response: The Results part illustrates the steps in our chemometrics workflow that allowed us to evaluate the very large dataset, shows findings from the different sample group comparisons, and demonstrates how the conclusions were reached. We included essential tables and figures that support the findings. Table 2 might appear relatively long, but it is already reduced to show only the critical results of the different statistical models used to compare the sample groups. The extensive underlying data and additional information can be found in the Supplementary material. Excepting the tables and figures, the text paragraphs in the Result part are rather short. The addition of new experiments and considerations would further increase the data amount to be assessed and probably not lead to add value to the existing information to such an extent that would make a manuscript extension favourable. We would thus keep the content of the Results as it is, but have made some changes that might improve readability. The Discussion closely follows the workflow described in the Results and aspires to find explanations for the findings, based on published references. We have tried to put the individual information in a greater context, and are convinced that the detected changes in the metabolite profiles are connected to progressing senescence in the spinach leaves during the storage period. Based on the shown significant differences in the PAW-treated material we were able to conclude that PAW can delay the senescence development in the fresh spinach.     

  1. Only describing a large amount of data is fairly preliminary and does not provide novel insights. The authors are recommended to follow up the literature about PAW treatment to find more interesting regulatory metabolites and for some major concern substances, quantitative or other tests should be added to further explain.

Response: The focus of the current project was on studying changes in the metabolite profiles of differently treated spinach leaves, and not on those of reactive oxygen and nitrogen metabolites in the plasma as performed by some of us in a recent study (Vaka et al. 2019, Foods 2019, 8, 692). We have therefore referenced our findings regarding the spinach metabolism to publications describing biochemical processes in detached leaves (references 28 to 62). With the aim to perform an unbiased analysis of the metabolite profiles, we chose to use an untargeted metabolomics approach, detecting as many metabolites as possible and not focussing on preselected ones as in quantitative determination.   The untargeted analysis in combination with extensive and stringent chemometrics permitted us to extract the most relevant and discriminating features from the large amount of data generated. We believe that the approach presented here is able to deliver valuable information with respect to changes in food quality and can be transferred to other foods.

--Minor points:

  1. Please check the use of ‘a/an’. Such as at Ln14, 58, 118, 209, 222, 317, 361 548 and 555.

Response: Indefinite articles (a/an) were added in the designated sentences.

2.Ln11, please change “which is a result of leaf senescence”.

Response:  The sentence has been modified.

3.Ln50-51。 The position of this sentence seems inappropriate.

Response:  The sentence is now in the correct place.

4.Ln78。"sregular"? Please revise this sentence.

Response: The word has been corrected

5.Ln94。Oxygen? Oyxgen?

Response: The word has been corrected.

6.Ln226。 Please add "were" after the sample. sotrage?

Response: The sentence has been corrected and modified.

7.Ln260-261。 Please revise this sentence.

Response: The sentence has been modified.

8.Ln287。 Is it relevant?

Response:  The information provided by the clusters is relevant as this revealed that several fragments, adducts, neutral losses or isomers, which are corresponding to one specific ion mass, were highly correlated. The sentence has been modified.

9.Ln293。 Statistically significant.

Response: The sentence has been modified.

  1. Please change the position of the legend in Figure 3.

Response: The position of the legend has been changes so that it is now below Figure 3.

11.Ln454。 Please add 'is' after and.

R: The sentence has been modified

12.Ln537。 Please delete "that".

R: The sentence has been corrected.

Reviewer 2 Report

The Review of Manuscript Number ID:  foods-1456290 titled:

Impact of plasma-activated water treatment on quality and shelf life of fresh spinach leaves evaluated by comprehensive metabolomic analysis

The manuscript contains very important issues regarding the problem of preserving the microbiological and sensory quality of minimally processed foods, such as fresh vegetable leaves, during storage. Fresh spinach leaves are highly nutritious but spoil very easily due to changes in microbiological quality and enzymatic browning as a result of leaf aging. Therefore, treatment with activated plasma (PAW) of spinach wash water can be very useful in ensuring food safety and extending shelf life.

The manuscript is very well prepared, it shows in-depth knowledge of the issues raised, both in terms of the research and analytical scope, as well as the discussion of the results. A comprehensive metabolomic analysis was performed to identify possible changes in the content of compounds present in the tissue of fresh and stored spinach leaves. High resolution mass spectrometry with liquid chromatography was used to compare metabolomes in control samples, tap water or PAW washed samples, followed by detailed statistics. No significant differences were found between the treatment groups at the beginning and the end of the storage period, but PAW has been shown to significantly increase the stress resistance and (antioxidant capacity) and aging delay of spinach leaves. Therefore, these studies may have the potential of treating PAV to improve the quality of other plant food raw materials and extend their shelf life. 

The manuscript is prepared properly and the issues are discussed extensively. It was based on 62 literature items, near 50% of which are from recent years (2017-2022), but old items were also used, even from 1879. It should be emphasized that the authors very synthetically presented issues that are closely related to the subject of the manuscript.

Author Response

Response: We appreciate the comments from the reviewer. Thank you also for highlighting the use of older references that can be valuable to understanding biochemical processes.

Reviewer 3 Report

Report on the submitted paper „Impact of plasma-activated water treatment on quality and shelf-life of fresh spinach leaves evaluated by comprehensive metabolomics analysis” by Rangel-Huerta et al for publication in foods (MDPI).

The research presents the results of studies aiming to investigate the influence of plasma-activated water (PAW) generated by non-thermal plasma (NTP) treatments on the metabolic activities of fresh spinach leaves. The research is of very high interest for the community working on the biological and biochemical effects of non-thermal plasmas (jets, DBDs and microwave), its generated PAW and plasma treated water (PTW). The insights of PAW with reactive oxygen and nitrogen species on different quality parameters of fresh green leaves like spinach, but other food research, too, is very important. Especially in the applied science, such research will improve the scientific understanding of mechanisms of action as well as the acceptance in industrial application.

The whole article is written in very good English. The article suits the formal standards and is well structured.

No additional comments.

Author Response

We would like to thank for the review and the feedback provided.

Round 2

Reviewer 1 Report

Although the author has revised some small points and explained major points, the key points are still not revised. The manuscript still lacks some novelty and interest.

Author Response

As stated by the reviewer, we have addressed several points that were considered relevant. The reviewer is of the opinion that the manuscript lacks novelty and interest, although the two other reviewers think otherwise. We consider that this work might be of interest for researchers in the field of food quality and metabolomics as we present an application that combines the power of HRMS and novel chemometrics tools that has the aim to enable an unbiased biological interpretation.